# UV-Cured Chitosan and Gelatin Hydrogels for the Removal of As(V) and Pb(II) from Water

**DOI:** 10.3390/polym14061268

**Published:** 2022-03-21

**Authors:** Camilla Noè, Michael Zanon, Amaya Arencibia, María-José López-Muñoz, Nieves Fernández de Paz, Paola Calza, Marco Sangermano

**Affiliations:** 1Politecnico di Torino, Dipartimento di Scienza Applicata e Tecnologia, C.so Duca Degli Abruzzi 24, 10129 Torino, Italy; camilla.noe@polito.it (C.N.); michael.zanon@polito.it (M.Z.); 2Departamento de Tecnología Química, Energética y Mecánica, ESCET, Universidad Rey Juan Carlos, C/Tulipán s/n, Móstoles, 28933 Madrid, Spain; amaya.arencibia@urjc.es; 3Departamento de Tecnología Química y Ambiental, ESCET, Universidad Rey Juan Carlos, C/Tulipán s/n, Móstoles, 28933 Madrid, Spain; mariajose.lopez@urjc.es; 4Department of Applications, Metrohm Hispania, C/Aguacate 15, 28044 Madrid, Spain; nieves.fernandez@metrohm.es; 5Dipartimento di Chimica, Università di Torino, Via P. Giuria 5, 10125 Torino, Italy; paola.calza@unito.it

**Keywords:** chitosan, gelatin, hydrogels, UV-curing, heavy metals adsorption

## Abstract

In this study, new photocurable biobased hydrogels deriving from chitosan and gelatin are designed and tested as sorbents for As(V) and Pb(II) removal from water. Those renewable materials were modified by a simple methacrylation reaction in order to make them light processable. The success of the reaction was evaluated by both ^1^H-NMR and FTIR spectroscopy. The reactivity of those formulations was subsequently investigated by a real-time photorheology test. The obtained hydrogels showed high swelling capability reaching up to 1200% in the case of methacrylated gelatin (GelMA). Subsequently, the Z-potential of the methacrylated chitosan (MCH) and GelMA was measured to correlate their electrostatic surface characteristics with their adsorption properties for As(V) and Pb(II). The pH of the solutions proved to have a huge influence on the As(V) and Pb(II) adsorption capacity of the obtained hydrogels. Furthermore, the effect of As(V) and Pb(II) initial concentration and contact time on the adsorption capability of MCH and GelMA were investigated and discussed. The MCH and GelMA hydrogels demonstrated to be promising sorbents for the removal of heavy metals from polluted waters.

## 1. Introduction

Considering the exponential growth of the industrial and urban areas affecting water quality, water contamination is becoming a global environmental issue that needs to be addressed urgently. Among the aqueous pollutants, heavy metals are considered the environmental priority contaminants due to their non-biodegradability, high toxicity, and bioaccumulative effects, which cause major negative effects on public health [1]. The most common heavy metals present in streams and lakes are arsenic, lead, copper, zinc, and nickel. The presence of these heavy metals in the environment can be addressed to both geological sources (volcanic emission and natural reaction) and anthropogenic sources (such as pharmaceutical, metallurgy, mining, electronic, and agricultural industries) [2].

Since the presence of those contaminants cannot be completely avoided the World Health Organization (WHO) has fixed their maximum amount in drinking water. For example, the concentrations of arsenic and lead should be less than 0.01 mg/L and 0.1 mg/L, respectively [3,4].

Different strategies have been employed for heavy-metal removals such as membrane filtration [5], photocatalysis [6] ion-exchange [7], coagulation [8], and adsorption [9]. Among them, adsorption is recognized as the most effective method for heavy metal polluted water remediation. Activated carbon, carbon nanotubes, mesoporous silica, and magnetic particles have been proposed in the literature as typical adsorbents for heavy metals removal from water [10,11,12]. However, those sorbents are usually quite expensive. To overcome this problem, different researchers have focused their attention on the development of new polymeric hydrogels (HG) as low-cost adsorbents. HG are three-dimensional polymer networks that can be physically or chemically crosslinked. They are particularly interesting compared to other adsorbent materials due to their water affinity, high swelling properties, and high porosity, which allows the diffusion of ions towards the polymeric network. In the last years, bio-derived sorbents have been recognized as promising candidates for water treatment. In fact, carbohydrate or other natural based-hydrogels can be interestingly exploited as bio-sorbents since they are renewable, bio-degradable, and non-toxic [13,14,15].

For this reason, cellulose, starch, chitosan, and alginate hydrogels have been recently proposed in environmental applications for the removal of heavy metals [16,17,18].

During the sorption process various hydrogel-heavy metal interactions can occur, such as electrostatic interaction, complexation, hydrogen bonding, and coordination/chelation depending on the type of pollutant, the experimental conditions (e.g., pH, temperature), ions concentration, and certainly on the different functional groups present on the polymeric hydrogels. Therefore, it is essential to properly design an adequate polymeric structure to remove the target heavy metal species in defined conditions. However, physically crosslinked natural hydrogels usually possess a low adsorption rate, poor stability, and poor mechanical properties. To overcome those drawbacks different strategies can be applied such as the chemical modification of the polymer or the development of hybrid hydrogels based on the mixture of organic and inorganic components [19,20,21].

Among biobased polymers, chitosan is one of the most promising. It derives from the *N*-deacetylation of chitin which is the second most abundant biopolymer in nature commonly present in fungi, algae, exoskeleton of crustacea, insects, and mollusks. The polymeric chains of chitosan are composed of a random mixture of β-(1→4)-d-glucosamine and *N*-acetyl-d-glucosamine units possessing many reactive groups such as amino and hydroxyl ones that can be easily modified [22]. Therefore, in recent years chitosan-based gels and membranes have attracted widespread research interest [23,24,25]. Another interesting natural polymer is gelatin which is an animal protein consisting of linear ionic polymeric chains bearing different amino and carboxylic functional groups. Gelatine is mostly used in biomedical applications due to its transparency and high biocompatibility [26,27].

In the past few decades, many attempts have been made to create chemically crosslinked hydrogels mainly involving the use of crosslinkers such as *N*,*N*′-methylenebisacrylamide [28], formaldehyde [29], and epichlorohydrin [30] which, nonetheless, are toxic and have to be removed from the hydrogels. Up to now, very few works have evaluated the possibility to modify biobased polymers to obtain photo-crosslinked hydrogels for water treatment and only two of them address specifically the removal of heavy metals [14,31,32].

Within this framework, we have methacrylated two natural polymers, chitosan, and gelatin, via a previously reported procedure [33,34]. This chemical modification makes these polymers suitable for photopolymerization in water, which is a fast-curing process and an environmentally friendly technique occurring at room temperature that allows achieving crosslinked hydrogels [35,36].

The modification of the starting biobased precursors was investigated by both ^1^H-NMR and FTIR spectroscopy. The curing process of the photocurable formulations was evaluated via a real-time photo-rheology. The swelling capability of the photocured hydrogels was investigated and correlated with their adsorption efficiency. Finally, the adsorption properties of the photocured hydrogels towards arsenic and lead species were deeply investigated by analyzing the equilibrium and kinetics of the adsorption process and the influence of pH.

## 2. Materials and Methods

### 2.1. Materials

Medium molecular weight chitosan (CH) (M_w_ = 190–310 KDa, 75–85% degree of *N*-deacetylation), methacrylic anhydride (MA), acetic acid (≥99%), Irgacure2959, gelatin from cold-water fish skin were purchased from Sigma-Aldrich (Milano, Italy) and used as received without further purification. Ammonium molybdate tetrahydrate [(NH_4_)_6_Mo_7_O_24_·4H_2_O (0.004M)], sulfuric acid [H_2_SO_4_ (1.25M)], potassium antimonyl tartrate trihydrate [C_8_H_4_K_2_O_12_Sb_2_·3H_2_O (0.004M)], ascorbic acid [C_6_H_8_O_6_ (0.06M)], sodium arsenate dibasic heptahydrate salt [Na_2_HAsO_4_·7H_2_O] and lead nitrate [Pb(NO_3_)_2_] were also purchased from Aldrich. All of the materials were used without further purification.

### 2.2. Synthesis of Methacrylated Chitosan (MCH)

The chitosan (CH) methacrylation was accomplished as previously reported [33]. Briefly, the CH (1.5 wt%) was solubilized in an acetic acid-water solution (2 wt%), then MA was added (molar ratio NH_2_:MA = 1:1). The mixture was then placed into the microwave furnace (Milestone STARTSynth, Milestone Inc., Shelton, CT, USA). The reaction time was set to 5 min, at 100 °C, and a launch time of 30 s. The obtained product was dialyzed for five days and subsequently freeze-dried.

### 2.3. Synthesis of Methacrylated Gelatin (GelMA)

The gelatin (Gel) methacrylation reaction was conducted by modifying a previously reported protocol [34]. Briefly, gelatin from fish skin was initially dissolved in distilled water at 50 °C (30 wt%). Then, MA was added dropwise (0.6 g of MA for 1 g of gelatin). The reaction was left to react for 4 h at 50 °C under stirring conditions. The pH of the solution was kept at 8 by adding NaOH solution (3 M). The product solution was dialyzed against distilled water for three days and then freeze-dried.

### 2.4. UV-Curing of Hydrogels

MCH (3 wt%) was solubilized in an acetic acid-water solution (2 wt%). Then, 2 phr (per hundred resin) of Irgacure 2959 was added to the solution as the photoinitiator. GelMA (10 wt%) was solubilized in distilled water, then 1 phr of Irgacure was added. Subsequently, the liquid solutions were poured into a silicon mold and irradiated with UV light (100 mW/cm^2^) using a Hamamatsu LC8 lamp equipped with an 8 mm light guide (240–400 nm as spectral distribution). The irradiation time was set at 5 min for the MCH formulation and 3 min for the GelMA one.

### 2.5. Characterization Techniques

#### 2.5.1. Proton Nuclear Magnetic Resonance (^1^H-NMR)

CH, MCH, Gel, and GelMA were analyzed by Bruker Advance 400 Fourier TransformNMRspectrometer (FT NMR, Bruker, Billerica, MA, USA) operating at 400 MHz. The ^1^H-NMR was conducted at room temperature. Approximately 8 mg of each sample were dissolved in 1 mL of D_2_O.

#### 2.5.2. Photorheology

The photorheology tests were performed with an Anton PAAR Modular Compact Rheometer (Physica MCR 302, Graz, Austria) using a parallel plate configuration (diameter = 15 mm) with a quartz bottom glass. The gap value was set as 300 μm. The time sweep experiment was performed in the linear viscoelastic region (LVR) at a constant strain amplitude (γ) of 0.5% and a constant frequency (ω) of 5 rad/s to monitor the in-situ gel formation by following the evolution of elastic storage modulus G’ with time. The reaction can be considered completed when the G’ plateau is reached. In these experiments, the UV lamp was switched on after 30 s. The UV lamp used in the photorheology experiments was a Hamamatsu LC8 lamp (Hamamatsu, Japan) with a light intensity of 28 mW/cm^2^. All experiments were carried out at room temperature and repeated in triplicates.

#### 2.5.3. Swelling

The swelling capability of the UV-cured hydrogels was tested by a gravimetric procedure. Air-dried samples were placed in distilled water at room temperature. The weight increase was monitored at different time steps by taking out the sample from water and weighting it after the removal of the surface free water. The swelling degree percentage (*SD*%), the swelling at equilibrium (*S_eq_*), and the equilibrium water content (*ECW*) were calculated with Equations (1)–(3), respectively.
(1)$$SD\%=\left(\frac{W_{t}-W_{d}}{W_{d}}\right)*100$$
(2)$$SD_{eq}=\frac{W_{e}-W_{d}}{W_{d}}$$
(3)$$EWC\%=\frac{W_{e}-W_{d}}{W_{e}}*100$$
where *W_t_* is the weight at time *t*, *W_d_* is the weight of the dry sample, and *W_e_* is the weight of the sample at the equilibrium state. All of the experiments were repeated in triplicates.

#### 2.5.4. Surface Charge

The electric charge on the hydrogels, zeta potential values, of samples were determined for aqueous particle suspensions using A NanoPlus DLS Zeta Potential from Micromeritics (Aqueous suspensions were prepared with a mass to volume ratio of 0.5 mg/mL at pH 2, 4, 6, and 9 by adjusting the pH with HCl and NaOH solutions.

#### 2.5.5. Adsorption Experiments

The adsorption experiments were performed to investigate the effect of Arsenic(V) and Lead(II) initial concentration, contact time, and pH by monitoring the decrease of As(V) and Pb(II) in the aqueous solutions. Each experiment was repeated three times, and the mean values were reported in this investigation.

The solutions of As(V) were prepared from sodium arsenate dibasic heptahydrate salt [Na_2_HAsO_4_·7H_2_O]. The As(V) concentration in the aqueous solution over time was assessed using a colorimetric procedure based on the formation of an arsenate-molybdate complex (max absorbance 884 nm) with a UV–visible spectrometer (JASCO V-630). This complex was formed by reacting As(V) with an acidic solution composed of ammonium molybdate tetrahydrate [(NH_4_)_6_Mo_7_O_24_·4H_2_O (0.004M)], sulfuric acid [H_2_SO_4_ (1.25M)], potassium antimonyl tartrate trihydrate [C_8_H_4_K_2_O_12_Sb_2_.3H_2_O (0.004M)] and ascorbic acid [C_6_H_8_O_6_ (0.06M)] [37].

The solution of Pb(II) was prepared from lead nitrate [Pb(NO_3_)_2_]. The total Pb concentration in solution was monitored by inductively coupled plasma atomic emission spectroscopy (ICP-AES) with a Varian Vista AX spectrometer, after calibration with stock solutions in the 0–15 mg/L range. Two emission mercury lines (217.00 and 220.35 nm) were used. The adsorbed amount was determined by the difference between initial and final concentrations in the solutions for each experiment.

To evaluate the effect of pH on the As(V) and Pb(II) adsorption four different solutions were prepared having the same initial concentration of metal ions but with pH = 2, 4, 6, and 9. The pH was adjusted with NaOH (3 M) and HCl (1 M) [38,39].

The adsorption kinetic study was conducted by contacting 15 mL of each metal ion solution (10–20 mg/L (As), 50–75 mg/L (Pb(II)), and 15 mg of the dried hydrogel. Subsequently, a fixed amount of supernatant was taken out at different time intervals to monitor the metal adsorption. On the other hand, the equilibrium adsorption isotherms were recorded by keeping the volume of the solution constant and varying the initial metal concentration in the 1–100 mg/L range for arsenic(V) with different individual points (1, 5, 10, 20, 30, 50, 75 and 100 mg/L respectively) while for Pb(II) the ion concentration was in the 1–200 mg/L range (10, 30, 50, 75, 100, 150, and 200 mg/L) maintaining the stirring for 24 h to assure equilibrium time. The experiments were conducted at 20 °C under mild stirring conditions.

The adsorbent capacity at time *t* (*q_t_* [mg/g]), the equilibrium adsorption capacity (*q_e_* [mg/g]) and the removal efficiency (*R%*) were calculated according to Equations (4)–(6), respectively [40,41].
(4)$$q_{t}=\frac{\left(C_{0}-C_{t}\right)\times V}{W}$$
(5)$$q_{e}=\frac{\left(C_{0}-C_{e}\right)\times V}{W}$$
(6)$$R\left(\%\right)=\left(\frac{C_{0}-C_{e}}{C_{0}}\right)\times 100$$
where *C*_0_ (mg/L) is the initial metal ion concentration, while *C_t_* (mg/L) and *C_e_* (mg/L) are the metal concentration at time t and at equilibrium, respectively. *V* (mL) is the volume of the metal solution, and *W* (g) is the mass of the dried hydrogel.

### 2.6. Adsorption Kinetics Models

Two different kinetic models were then used to evaluate the adsorption rate and the potential rate-controlling step. The kinetic data were analyzed by means of pseudo-first-order and pseudo-second-order models [42], using the Lagergren Equations (7) and (8).
(7)$$\frac{dq_{t}}{dt}=k_{1}\left(q_{e}-q_{t}\right)$$
(8)$$-\ln\left(1-\frac{q_{t}}{q_{e}}\right)=k_{1}t$$
where *k*_1_ is the rate constant of pseudo-first-order sorption [1/min]. According to this approximation, a plot of −ln(1 − (*q_t_/q_e_*)) vs. *t* gives a straight line with slope *k*_1_.

Equations (9) and (10) report the second-order kinetic rate equation and its integrated formula, respectively.
(9)$$\frac{dq}{dt}=k_{2}{\left(q_{e}-q_{t}\right)}^{2}$$
(10)$$\frac{t}{q_{t}}=\frac{1}{k_{2}q_{e}^{2}}+\frac{t}{q_{e}}$$
where *k*_2_ is the rate constant of the pseudo-second-order sorption [g/(mg ∗ min)]. According to this approximation, a plot of *t/qt* vs. *t* gives a linear relationship with slope 1/*q_e_* and intercept 1/*k*_2_*qe*^2^.

### 2.7. Equilibrium Isotherms Models

Three equilibrium isotherm models, Langmuir, Freundlich, and Sips were used to describe the adsorption mechanism.

The Langmuir model (Equation (11)) considers the adsorption to be chemisorption and can be applied to homogeneous adsorption phenomena, in which the metal adsorption energy is constant through every site of the surface, thus explaining the formation of a monolayer of adsorbate [43].
(11)$$q_{e}=\frac{q_{m}K_{L}C_{e}}{1+K_{L}C_{e}}$$
where *C_e_* (mg/L) is the equilibrium concentration of adsorbate in the remaining solution, *q_m_* (mg/g) is the adsorbed amount present in the monolayer related to the maximum adsorption capacity, *K_L_* (L/mg) is Langmuir constant related to the metal affinity to the binding sites.

The Freundlich model (Equation (12)) can be applied to the heterogeneous surface with different energies for monolayer surface adsorption or to the formation of multilayers of adsorbate. The model can be expressed by the following equation:(12)qe=KFCe1n
where *Ce* (mg/L) is the equilibrium concentration of adsorbate in the remaining solution, and K_F_ and n are the empirical Freundlich constant and the heterogeneity factor, respectively. The value *1/n* indicates if the isotherm is favourable (0 < 1/*n* < 1), unfavourable (1/*n* > 1) or irreversible (1/*n* = 0) [44].

Sips model (Equation (13)) is a hybrid model which combines the Langmuir and the Freundlich models. This model is able to describe the homogeneous or heterogeneous model for monolayer adsorption. The Equation (12) represents the non-linear Sips isotherm.
(13)$$q_{e}=\frac{q_{m}K_{s}C_{e}^{n_{s}}}{1+K_{s}C_{e}^{n_{s}}}$$
where *q_m_* (mg/g) is the maximum adsorbed amount, *K_S_* is the Sips constant related with the affinity between the metal and the adsorption site and *n_s_* is the Sips exponent (dimensionless). It can be noticed that the Sips model becomes the Langmuir model when *n_s_* = 1, and the Freundlich model at low *C*_0_ [45,46,47].

## 3. Results and Discussion

### 3.1. Bio-Based Polymers Methacrylation

Chitosan and gelatin were methacrylated, following the previously reported experimental procedure, to make them photocurable [33,34]. The schemes of the methacrylation reactions are reported in Figure 1.

The CH and Gel methacrylation reactions were investigated and confirmed by ^1^H-NMR and FTIR. Figure 2a shows a comparison between the CH and MCH ^1^H-NMR spectra. The CH spectrum displays the typical CH peaks: the quadruplet peak at δ = 3.59 ppm represents the protons in 1–4, 6–10, 12 positions while the peak at 3.02 ppm represents the protons in 5, 11 positions of the chitosan ring. The peak at δ = 1.9 ppm represents the (*N*-acetyl)d-glucosamine group [48,49]. In the MCH spectrum, new peaks can be seen at δ = 6, 5.60 and 5.41 ppm representing the =CH_2_ of the methacrylic double bonds and at δ = 1.85 and 1.78 ppm corresponding to the −CH_3_ methyl groups of the grafted methacrylated moieties. As can be observed, there are two types of −CH_3_ signals and three different peaks corresponding to the =CH_2_ protons, meaning that the methacrylated groups are bonded to both the −NH_2_ and the −OH groups of the chitosan.

From the integrals of the ^1^H-NMR of MCH, it was possible to calculate the degree of substitution (*DOS*) following Equation (14).
(14)$$DOS=\left(\frac{\frac{I_{6}+I_{5.6}+I_{5.41}}{2}}{\frac{I_{3.59}+I_{3.02}}{14}}\right)/6$$
where *I*_6,_
*I*_5.6_ and *I*_5.41_ are the integrals of the =CH_2_ intensity, coloured violet on the spectrum, and the *I*_3.59_ and *I*_3.02_ are the integrals of the H protons in 1–12 positions, coloured in green and blue. The whole formula is divided by six since there are approximately six reactive groups in each chitosan double rings (not taking into account the acetylation side chain). The obtained DOS was 0.27, which is very similar to the one previously reported by other methacrylated chitosan [48,50].

In Figure 2b are reported the Gel and GelMA ^1^H-NMR spectra in which the success of the methacrylation reaction was clearly assessed. In fact, in the GelMA spectrum, it is possible to observe the presence of new peaks at δ = 5.6 and 5.8 ppm representing the =CH_2_ protons and at δ = 3.5 ppm which can be ascribed to the CH_3_ protons of the methacrylated group [48,51,52]. However, since the complete gelatin structure is still unknown, due to the presence in its chains of a vast variety of amino sequencies, it was not possible to calculate the GelMA degree of substitution.

The methacrylation reactions were further confirmed by FTIR analysis. In Figure 3a are reported the FTIR spectra of CH and MCH. In the spectrum of MCH there can be clearly observed the presence of a new peak at 1720 (1/cm) that can be attributed to the C=O stretching vibrations, and at 1620 and 810 (1/cm) which can be attributed to the C=C and C=CH_2_ stretching and out of plane bending vibrations, respectively [49,53,54,55]. In the spectra comparison, it can also be observed a shift of the peak centred in 1580 (1/cm) towards lower a wavenumber (1538 (1/cm)) assigned to the −NH stretching vibration indicating a *N*-methacrylation [50]. Moreover, it can also be observed a decrease of the broad band centred in 3300 (1/cm) assigned to the −OH vibrations, suggesting that also the hydroxyl groups can be a grafting site for the methacrylated group, which is in good agreement with the ^1^H-NMR result.

In Figure 3b the spectra of Gel before and after the methacrylation reaction are reported. Also in this case, the accomplishment of the methacrylation reaction was confirmed by the appearance of new bands in the GelMA spectrum at 1380 and 830 (1/cm) which can be assigned to the C−O stretching and C=C bending vibrations, respectively [56,57].

### 3.2. Photoreactivity of Methacrylated Polymers and Swelling of UV-Cured Hydrogels

The photoreactivity of the methacrylated polymers dispersed into deionized water was investigated by means of real-time photorheology. Either the MCH and GelMA formulations were investigated.

The phothorheology curves of the MCH and GelMA formulations are reported in Figure 4. As it can be observed, the MCH formulation started reacting immediately after the lamp was switched on and reached a G’ plateau after 300 s. On the contrary, the GelMA formulation showed an induction time, i.e., the minimum time required to start the photocrosslinking, of about 20 s and reached a G’ plateau after 180 s.

These data clearly indicate a very high reactivity of the methacrylated polymers towards the radical-induced UV-curing process, leading to the formation of crosslinked hydrogel networks.

The swelling capability of the UV-cured hydrogels was measured following the experimental procedure previously described. The swelling curve of the UV-cured hydrogels is reported in Figure 5, while the *SD_eq_* and *EWC* are reported in Table 1. Interestingly, as can be observed from Figure 5, GelMA hydrogel shows superior swelling capability with respect to the MCH ones, with a final swelling degree of about 1200%. Besides the different molecular structure of those two polymers, the huge differences in the final swelling degree can possibly be ascribed to a lower GelMA functionalization with respect to MCH leading to a lower crosslinking density and resulting in the decrease of the swelling capability of the hydrogel in the aqueous solution [15,58]. The high equilibrium water content (*EWC*%) values measured from both hydrogels assessed their high-water permeability [59].

### 3.3. Removal of As(V) and Pb(II) from Water

The influence of pH variation toward hydrogel adsorption capacity was monitored for both metals as the ionic state of the surface functional groups on hydrogel can change. On MCH and GelMA hydrogels, surface free amino-groups (−NH_2_) are still present after the methacrylation reaction. Chitosan has primary amino groups and a pK_a_ value close to 6.5 [58]. Therefore, at pH lower than 6.5 the free-amino groups become protonated, inducing an enhancement of positive charges on the surface, which provokes a repulsion among the polymeric chains. On the contrary, in alkaline conditions, the amine groups are completely deprotonated, which results in the attraction of polymer chains due to Van der Waals forces. These aspects together with the metal charge are of paramount importance to unravel the adsorption process [60].

#### 3.3.1. As(V) Adsorption from Water

The As(V) adsorption was initially investigated by varying the pH from 2 to 9. Note that no degradation was observed when the samples were immersed in the solutions at different pH.

The Z-potential of the hydrogels was evaluated at different pH and correlated with the *q_e_* values measured.

From the experimental data collected in Figure 6a comes up that the optimum condition for the As(V) adsorption with MCH was achieved at pH = 6. In fact, a favourable interaction between the positive charged surface of the hydrogel and the negatively charged metal may be occurring; furthermore, at this pH a balance between the charge of the surface amino groups and the distance between the polymer chains it can exist.

Regarding the charge of the surface, zeta potential measured for MCH shows that the surface is positively charged from pH 2 to pH 6, with a point of zero charge (PZC) of ca. 7.4. GelMA surface displays also a positive Z-potential at low pH, but the PZC is close to 4.4. To explain the effectiveness of arsenic adsorption, its speciation in solution must be taken into account. The equilibrium dissociation constants indicate that arsenate species are negative at pH > 2.2 (Equations (15)–(17)). Therefore, As(V) was better adsorbed on the positive surface of MCH at pH 4 and 6, where H_2_AsO_4_^−^ species is predominant, while the adsorption decreased for pH 9 since the surface of the hydrogel changes to negative.
H_3_AsO_4_ → H_2_AsO_4_^−^ + H^+^ pKa_1_ = 2.20 (15)
H_2_AsO_4_^−^ → HAsO_4_^2−^ + H^+^ pKa_2_ = 6.94(16)
HAsO_4_ ^2−^ → AsO_4_^3−^ + H^+^ pKa_3_ = 11.50(17)

However, even if the GelMA surface shows a positive Z-potential at low pH, the overall As(V) adsorption is quite low. This result can possibly be ascribed to the presence of a high number of carboxylic groups in the GelMA structure which can interact with the amine also present in its structure, hindering the As(V) adsorption capability [61,62].

Figure 7 reports the adsorption kinetics profiles of As(V) obtained for two initial concentration (10 and 20 mg/L) at different time intervals at pH 6. The adsorption on the hydrogels is fast with GelMA as it reaches a plateau after 1 h, while in the case of MCH 8h are required (see Figure 7a,d). In both cases, the adsorption rates are higher in the initial part of the experiments suggesting that the adsorption mainly occurs on the surface of the hydrogels. The MCH sorption equilibrium time is slightly shorter than the one previously reported in the literature for other chitosan-based systems such as chitosan beads (24 h) [38] and modified chitosan gel beads (24 h) [63].

The adsorption equilibrium *q_e_* is reached slightly later when the initial As(V) concentration was set to 20 mg/L (Figure 7c).

Two mathematical models were used to fit the data and to perceive the mechanism of adsorption. The pseudo-first-order model resulted not applicable as the obtained trend is not linear (Figure 7b,e). On the contrary, a linear trend was obtained when data were plotted by applying the pseudo-second-order kinetics model (Figure 7c,f) which means that this model can correctly describe the mechanism for the As(V) adsorption. According to this model, the main adsorption mechanism is chemisorption. The pseudo-second-order fitting parameters at two different initial As(V) concentrations are reported in Table 2.

Noteworthy, the removal percentage [R(%)] of MCH was 98 and 92% for *C*_0_ = 1 and 5 mg/L, respectively, indicating a high As(V) removal efficiency of this hydrogel at low concentration.

Figure 8 shows the experimental data of equilibrium adsorption of As(V) along with their non-linear isotherm models used to fit the adsorption data (Langmuir, Freundlich and Sips). The parameters of the adsorption isotherm model are summarized in Table 3. The isotherm model with the highest R^2^ value was selected as the most representative of the As(V) adsorption process. All of the hydrogels adsorption mechanisms can be correctly described by the Sips model.

For both MCH and GelMA the value 1/n is less than 1 in the Langmuir model, so implying that the adsorption of As(V) metal is favourable. Furthermore, the maximum q_e_ values obtained for the adsorption of the As(V) ions are slightly higher than the one previously reported in the literature using other types of chitosan and other adsorbents such as goethite and cellulose (see Table 5).

#### 3.3.2. Lead Adsorption from Water

The adosorption capacity of Pb(II) was investigated following the same procedure previously reported for As(V) oxyanions. Figure 9 reports the Pb(II) adsorption of the MCH and GelMA hydrogels as a function of pH. At pH 6 the GelMA, with the net negative surface charge is the only material capable to interact with the positively charged metal cation. At basic pH it can be noticed that both MCH and GelMA possess a negative Z-potential and can therefore be applied to lead removal (see Figure 6b). As such, all of the Pb(II) adsorption tests were conducted only on MCH at pH = 9 and on GelMA at pH = 6.

In Figure 10a,d are reported the Pb(II) adsorption kinetics of MCH at pH = 9 and GelMA at pH = 6 at *C*_0_ = 50 mg/L and *C*_0_ = 75 mg/L, respectively, while Table 4 collects the kinetic fitting parameters. As in the case of As(V), the first-order kinetics model is not suitable to describe the kinetics profile of MCH and GelMA hydrogels, while the second-order kinetic model could fit the experimental values of Pb(II) adsorption; therefore, again the rate-determining step of the process is chemisorption.

Figure 11a shows the isotherm values obtained for MCH and GelMA hydrogels, where it can be seen that both solids are able to adsorb significant amounts of Pb(II), achieving values of ca 66 and 48 mg/g, respectively. It is important to underline that the achieved maximum values of Pb(II) adsorption capacities are similar to those previously obtained with other adsorbents. For comparison purpose, Table 5 summarizes the list of other Pb(II) absorbents available in the literature. Figure 11b, reports the R(%) of MCH and GelMA at the different initial Pb(II) concentrations. Interestingly, those percentages were very high, achieving almost 99% for low initial concentration and (>80%), also with high content of metal, so highlighting the great removal capability toward Pb(II) of these biobased hydrogels.

## 4. Conclusions

New biobased UV-curable hydrogels were developed using modified chitosan and gelatin and applied as adsorbents for the removal of As(V) and Pb(II) from an aqueous solution. Methacrylated chitosan (MCH) and methacrylated gelatin (GelMA) were successfully synthetized as assessed by ^1^H-NMR and FTIR spectroscopy. The modified materials were then dispersed in water together with BAPO photoinitiator. The high reactivity of these formulations was confirmed by a photo-rheology test, in which the rapid in situ gel formation was evaluated by following the evolution of elastic storage modulus G’ with time. MCH formulation reached a G’ plateau after 300 s, while the GelMA formulation required only 180 s. The swelling test highlights that those hydrogels possess a very high swelling capability reaching 616% and 1230% for MCH and GelMA, respectively. The adsorption parameters such as initial metal ions concentration, contact time, and pH noticeably influenced the removal efficiency of the hydrogels. The Z-potential analysis was performed on the hydrogels at different pH and the outcomes were used to discuss the measured q_e_ values of the hydrogels. The kinetics studies revealed that a pseudo-second order kinetic model can correctly describes the adsorption of As(V) and Pb(II) on MCH GelMA hydrogels, therefore suggesting a chemical adsorption. The obtained q_e_ were in the same order of magnitude than one previously reported in the literature for other types of chitosan and other adsorbents such as goethite and cellulose. All o the As(V) isotherms were correctly fitted by the Sips model. The removal efficiency of As(V) in water was very high for MCH reaching up to 98 and 92% in the case of *C*_0_ = 1 and 5 mg/L. Even better results were reached for the Pb(II) removal in which the R(%) remains above 80% for both MCH and GelMA even for *C*_0_ = 75 mg/L.

To conclude, this study successfully demonstrates the possibility to use modified chitosan and gelatin to obtain innovative UV-curable sorbents for the removal of As(V) and Pb(II) from polluted waters.

## Figures and Tables

**Figure 1 polymers-14-01268-f001:**
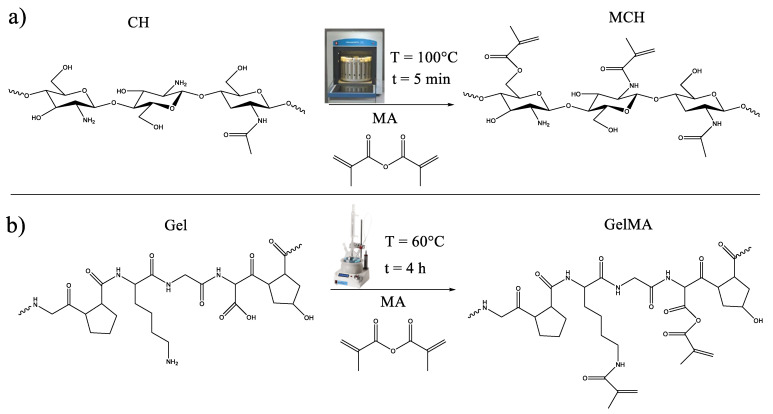
Scheme of the methacrylation reaction of (**a**) chitosan and (**b**) gelatin.

**Figure 2 polymers-14-01268-f002:**
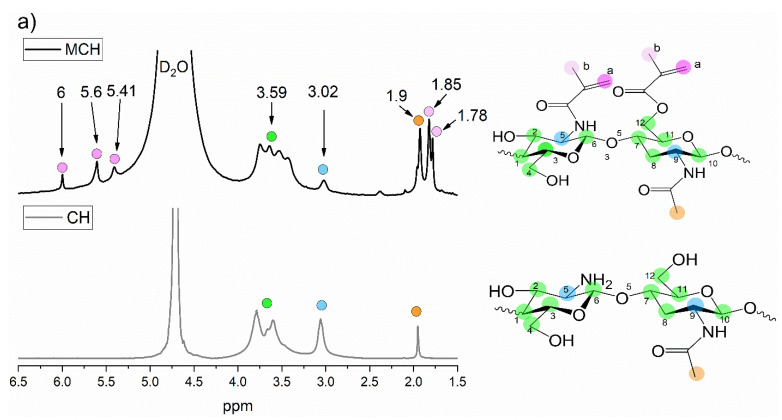
^1^H-NMR spectra of (**a**) CH and (**b**) Gel before and after the methacrylation reaction.

**Figure 3 polymers-14-01268-f003:**
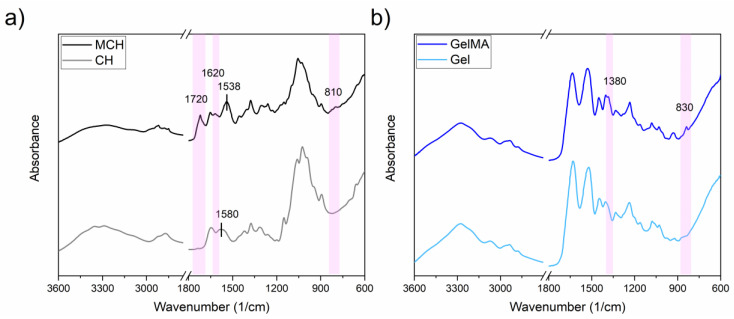
FTIR spectra of (**a**) CH and (**b**) Gel before and after the methacrylation reaction.

**Figure 4 polymers-14-01268-f004:**
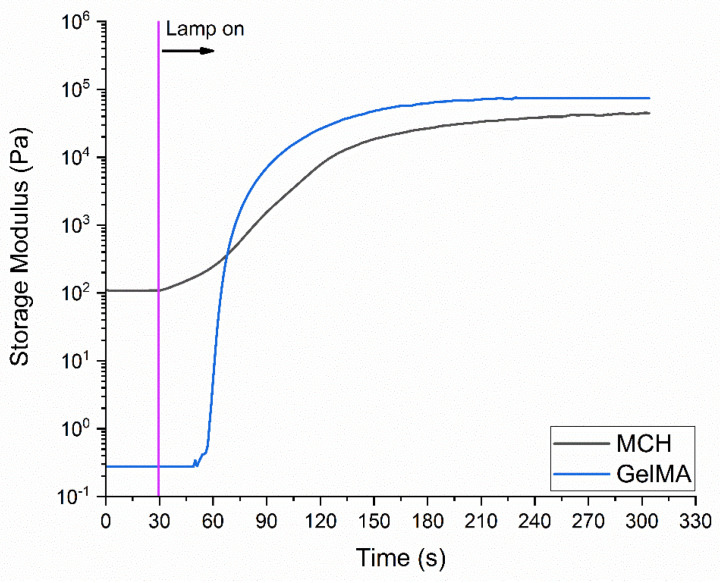
Photorheology curves recorded for the MCH formulation (3 wt% solubilized in acetic acid-water solution containing 2 phr of Irgacure 2959), GelMA formulation (10 wt% solubilized in distilled water containing 1 phr of Irgacure 2959).

**Figure 5 polymers-14-01268-f005:**
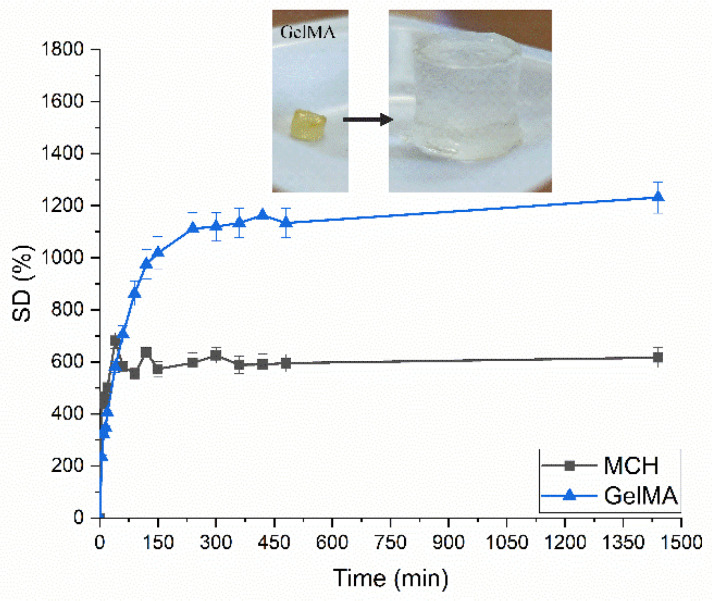
Swelling curves for UV-Cured MCH and GelMA hydrogels. In the insert is reported the swelling of GelMA UV-Cured hydrogels as an example.

**Figure 6 polymers-14-01268-f006:**
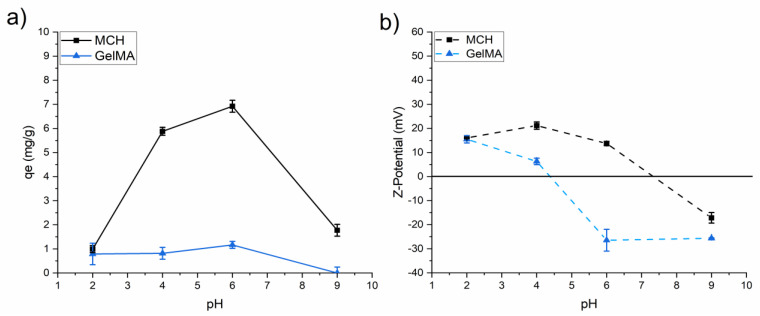
(**a**) *q_e_* values of As(V) as a function of pH (*C*_0_ = 10 mg/L) and (**b**) Z-potential as a function of pH for the different investigated UV-Cured hydrogels.

**Figure 7 polymers-14-01268-f007:**
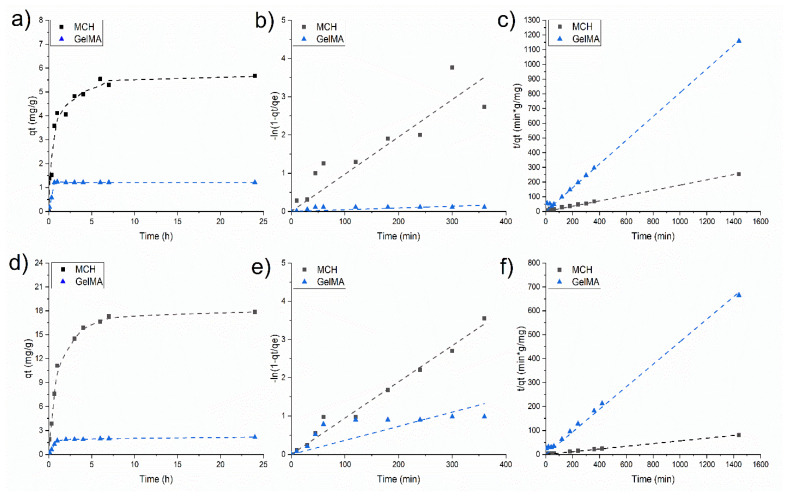
As(V) adsorption kinetics at (**a**) 10 mg/L and (**d**) 20 mg/L of initial concentration; (**b**,**e**) data fitted using the pseudo-first-order kinetic model with 10 mg/L and 20 mg/L of *C*_0_, respectively (**c**,**f**) data fitted using the pseudo-second-order kinetic model with 10 mg/L and 20 mg/L of *C*_0_, respectively.

**Figure 8 polymers-14-01268-f008:**
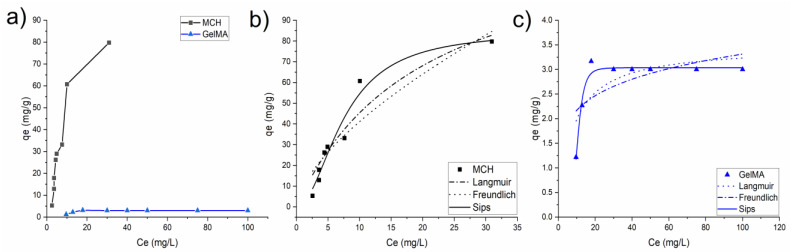
(**a**) Isotherm plots of MCH and GelMA hydrogels obtained from the experimental data; (**b**) MCH and (**c**) GelMA isotherm values fitted with different isotherm models.

**Figure 9 polymers-14-01268-f009:**
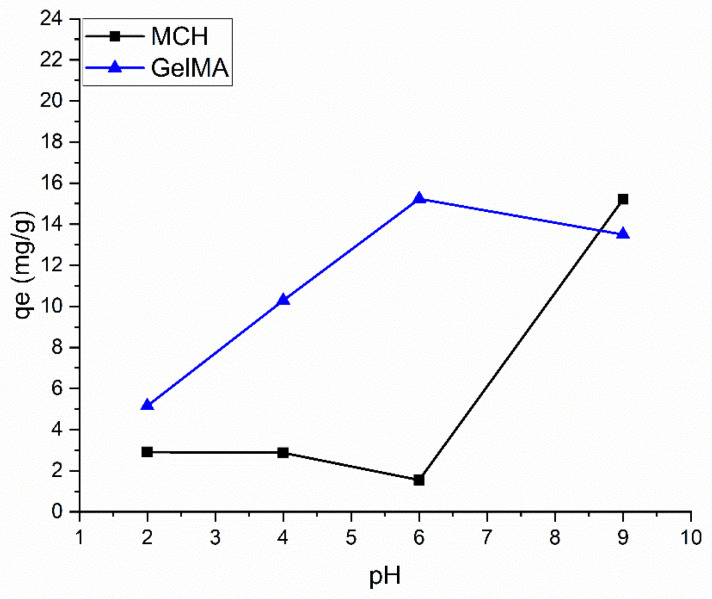
Effect of pH on hydrogel Pb adsorption. Those data were recorded with 30 mg/L Pb initial concentration.

**Figure 10 polymers-14-01268-f010:**
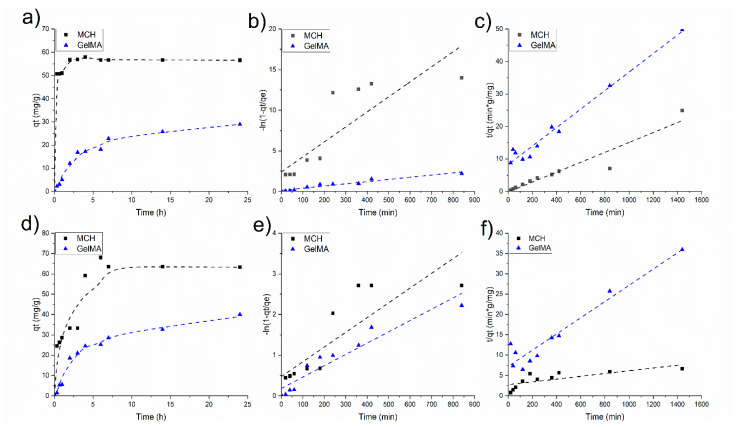
Pb(II) adsorption kinetics at (**a**) 50 mg/L and (**d**) 75 mg/L of initial concentration; (**b**,**e**) data fitted using the pseudo-first-order kinetic model with 50 mg/L and 75 mg/L of *C*_0_, respectively, (**c**,**f**) data fitted using the pseudo-second-order kinetic model with 50 mg/L and 75 mg/L of *C*_0_, respectively.

**Figure 11 polymers-14-01268-f011:**
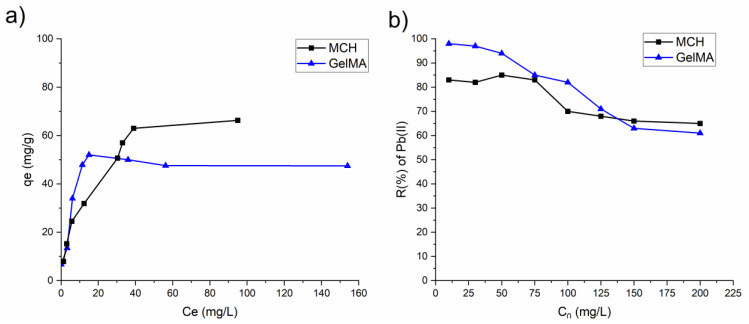
(**a**) Isotherm plots of MCH and GelMA hydrogels obtained from the experimental data; (**b**) removal percentage R(%) of Pb(II) by MCH and GelMA at different initial concentration.

**Table 1 polymers-14-01268-t001:** Swelling properties of UV-Cured hydrogels.

	*SD_eq_*	*EWC*%
MCH	6.16 ± 0.4	86 ± 3
GelMA	12.31 ± 0.7	93 ± 1

**Table 2 polymers-14-01268-t002:** Pseudo-second order fitting parameters for As(V) adsorption.

***C*_0_(As) = 10 mg/L**				
	*q_e,calc_* [mg/g]	*q_e,exp_* [mg/g]	*k*_2_ [g/(mg × min)]	R^2^
**MCH**	5.67	5.59	0.18	0.994
**GelMA**	1.22	1.24	0.81	0.998
***C*_0_(As) = 20 mg/L**				
	*q_e,calc_* [mg/g]	*q_e,exp_* [mg/g]	*k*_2_ [g/(mg × min)]	R^2^
**MCH**	17.85	17.54	0.06	0.994
**GelMA**	2.17	2.13	0.47	0.996

**Table 3 polymers-14-01268-t003:** Adsorption isotherm parameters for As(V) adsorption on MCH and GelMA at pH = 6.

**Langmuir parameters**				
	*q_m_* [mg/g]	*K_L_* [L/mg]	R^2^	
**MCH**	136.7	0.05	0.885	
**GelMA**	3.5	0.13	0.543	
**Freundlich parameters**				
	*n*	*K_F_* [L/mg]	R^2^	
**MCH**	1.56	9.44	0.831	
**GelMA**	5.47	1.42	0.339	
**Sips parameters**				
	*q_m_* [mg/g]	*n_s_*	*K_S_* [L/mg]	R^2^
**MCH**	84.8	2.0	0.02	0.939
**GelMA**	3.0	6.0	8.59	0.963

**Table 4 polymers-14-01268-t004:** Pseudo-second order fitting parameters for Pb(II) adsorption.

***C*_0_(Pb) = 50 mg/L**				
	*q_e,calc_* [mg/g]	*q_e,exp_* [mg/g]	*k*_1_ [g/(mg × min)]	R^2^
**MCH**	56.55	55.56	0.02	0.897
**GelMA**	28.94	33.33	0.03	0.975
***C*_0_(Pb) = 75 mg/L**				
	*q_e,calc_* [mg/g]	*q_e,exp_* [mg/g]	*k*_2_ [g/(mg × min)]	R^2^
**MCH**	63.26	700	0.003	0.685
**GelMA**	40.05	50	0.02	0.985

**Table 5 polymers-14-01268-t005:** Maximum As(V) and Pb(II) equilibrium adsorption capacity of different adsorbents reported in the literature.

Adsorbent	Metal	Initial Concentration-*C*_0_ [mg/L]	Temperature [°C]	Equilibrium Adsorption Capacity *q_e_* [mg/g]	References
Goethite	As(V)	1000	25	12	[64]
Chitosan		400	25	58	[63]
Cellulose with Fe_2_O_3_		100	25	32	[65]
Ce–Fe oxide decorated multiwalled carbon nanotubes		20	25	31	[66]
MCH GelMA		100	25	36 22	Present work
Chitosan/PVA	Pb(II)	30	25	0.9	[67]
Chitosan		250	25	47.4	[68]
Chitosan(Chitin)/Cellulose Composite		160	25	27.31	[69]
MCH GelMA		150	25	66 48	Present work

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
