# Peer review of "UV-Cured Chitosan and Gelatin Hydrogels for the Removal of As(V) and Pb(II) from Water"

_polymers, 2022, doi:10.3390/polym14061268_

Round 1

Reviewer 1 Report

The manuscript is acceptable and all the results are explained as well. 

No comment to the authors.

Author Response

The answer to the reviewer is in the attached file

Reviewer 2 Report

The manuscript is interesting and clear. It merits publication.

Minor remark: Unless I missed it, the instrumentation and methodology for 1H NMR and zeta-potential characterization is not given in section 2.5.

Author Response

The answer to the reviwer is in the attached file

Reviewer 3 Report

The authors present an interesting paper using two different hydrogels to remove two common heavy metal pollutants from water. The manuscript presents detailed data, which is used discusses reaction kinetics and isotherm models for chemically well-characterized substances.

One correction of the contents should be considered:
l.349 The R2 criterion shows that the Sips model is most representative, within which the parameter n belongs to the Freundlich part of the model. Please correct.

Further, a few minor typing errors shoudl be corrected:
238/239 "towards a lower wavenumber"
233-243 check comma placement
252 "Both, the MCH and GelMA..."

Author Response

(The authors gave the same response as above.)

Reviewer 4 Report

1) The introduction is weak and does not provide sufficient background information and context to the presented research.

2) The NMR spectra in Figure 2 should be annotated. Peak integration and peak picking with annotation should be presented.

3) Error bars are provided, which is appreciated. However, their derivation, the number of independently prepared materials should be mentioned in each figure caption.

4) Some data on pH dependence is provided, however, the stability limits are not discussed in the manuscript.

5) More adsorption isotherm fittings should be presented, such as Pedersen and Toth, and the discussions should be more in-depth.

6) The selectivity and matrix effect should be discussed in details with real samples at real concentrations.

7) Chitosan in materials science is emerging with widespread applications that should be briefly mentioned and exemplified (10.1021/acssuschemeng.1c07047; 10.1021/acssuschemeng.1c06786; 10.1039/D1GC02679H).

8) The purity of all chemicals, materials and solvents should be listed under the experimental section of the manuscript for reproducibility purposes.

9) Both the quotient (“x/y”) and negative exponent (“x y-1”) formats are used in the manuscript for units. Either of them should be used consistently, preferably the negative exponent format, which is recommended by the IUPAC.

Author Response

The answer is the attached file

Round 2

Reviewer 4 Report

Comments have been addressed. Manuscript is acceptable.